

# Genome-wide analysis, identification, evolution and genomic organization of dehydration responsive element-binding (DREB) gene family in *Solanum tuberosum*

Nida Mushtaq[1], Faiza Munir[1], Alvina Gul[1], Rabia Amir[1] and Rehan Zafar Paracha[2]

[1] Department of Plant Biotechnology, Atta-ur-Rahman School of Applied Biosciences, National University of Sciences and Technology, Islamabad, Pakistan
[2] Research Centre for Modelling & Simulation, National University of Sciences and Technology, Islamabad, Pakistan

## ABSTRACT

**Background:** The dehydration responsive element-binding (DREB) gene family plays a crucial role as transcription regulators and enhances plant tolerance to abiotic stresses. Although the DREB gene family has been identified and characterized in many plants, knowledge about it in *Solanum tuberosum* (Potato) is limited.
**Results:** In the present study, StDREB gene family was comprehensively analyzed using bioinformatics approaches. We identified 66 StDREB genes through genome wide screening of the Potato genome based on the AP2 domain architecture and amino acid conservation analysis (Valine at position 14th). Phylogenetic analysis divided them into six distinct subgroups (A1–A6). The categorization of StDREB genes into six subgroups was further supported by gene structure and conserved motif analysis. Potato DREB genes were found to be distributed unevenly across 12 chromosomes. Gene duplication proved that StDREB genes experienced tandem and segmental duplication events which led to the expansion of the gene family. The Ka/Ks ratios of the orthologous pairs also demonstrated the StDREB genes were under strong purification selection in the course of evolution. Interspecies synteny analysis revealed 45 and 36 StDREB genes were orthologous to *Arabidopsis* and *Solanum lycopersicum*, respectively. Moreover, subcellular localization indicated that StDREB genes were predominantly located within the nucleus and the StDREB family's major function was DNA binding according to gene ontology (GO) annotation.
**Conclusions:** This study provides a comprehensive and systematic understanding of precise molecular mechanism and functional characterization of StDREB genes in abiotic stress responses and will lead to improvement in *Solanum tuberosum*.

# INTRODUCTION

Dehydration responsive element binding (DREB) is considered as one of the largest and best studied gene family involved in the abiotic stress mitigation by regulating the expression of genes involved in ABA-independent stress tolerance pathway (*Lata &*

Corresponding author
Faiza Munir,
faiza.munir@asab.nust.edu.pk

*Prasad, 2011*). DREB belongs to the AP2 multigene family and possess a single conserved AP2 domain (*Dietz, Vogel & Viehhauser, 2010*). The DREB gene family is characterized by the presence of valine (V) at position 14th and glutamic acid (E) at position 19th respectively within the conserved AP2 domain (*Zhou et al., 2010*). The AP2 domain consists of 60–70 conserved amino acid residues, indispensable for plant's stress and defense response mechanisms (*Wu et al., 2015*). The AP2 domain comprises seven key amino acids that are necessary for DRE binding: one V residue, four R residues, and two W residues (*Allen et al., 1998*). In addition, a conserved Serine/Threonine rich region is located adjacent to the AP2 domain, which is responsible for phosphorylation of the DREB genes (*Shen et al., 2003*). Based on their structural characteristics, DREB genes can be further divided into six subgroups: A1–A6 (*Zhou et al., 2010*). DREB transcription factors are one of the most promising regulons for abiotic stress tolerance in plants that directly interact with DRE/CRT, which consist of core motifs ACCGAC/GCCGAC (*Sharoni et al., 2011*; *Vazquez-Hernandez et al., 2017*). These sequences are found in the promoter regions of drought and cold responsive genes for example KIN1 and KIN2 (cold responsive) (*Kimotho, Baillo & Zhang, 2019*), RD29A (drought responsive) (*Zhao et al., 2013*), LEA (Late Embryogenesis Abundant) (*Liang et al., 2019*), COR15A and COR15B (cold responsive) (*Yu, Wang & Zhang, 2018*).

DREB gene family members are widely involved in abiotic stress responses such as drought, salt, cold, and heat (*Du et al., 2018*; *Kudo et al., 2017*; *Wu et al., 2018*). DREB genes were first identified in *Arabidopsis*. AtDREB1 and AtDREB2 can function as two independent proteins in two distinct signal transduction pathways under cold and dehydration stress, respectively (*Liu et al., 1998*). In *Arabidopsis*, studies have shown that ABA signaling does not normally mediate the expression of the A1 and A2 subgroups. On the contrary, ABI4, the only member of A3 subgroup is involved in sugar and ABA signaling (*Shkolnik-Inbar & Bar-Zvi, 2011*). A4 subgroup includes the most studied members namely TINY which functions in drought tolerance and HARDY which participates in both salinity and drought tolerance (*Xie et al., 2019*). Drought and cold stress induce RAP2.1, member of A5 subgroup, in *Arabidopsis* (*Dong & Liu, 2010*). RAP2.4 and RAP2.4B genes in the A6 subgroup are responsive to drought, heat and salinity stress respectively (*Rae, Lao & Kavanagh, 2011*). DREB genes are reported to impart enhanced drought tolerance in many plant species. Overexpression of DREB1A led to heightened drought tolerance in *Arabidopsis* and tobacco. A similar phenotype is reported in wheat using the RD29A promoter. Likewise, DREB gene from cotton conferred enhanced tolerance to drought, heat and cold stress when expressed in wheat (*Rabara, Tripathi & Rushton, 2014*). Constitutive overexpression of CsDREB gene led to enhanced drought and salt tolerance in transgenic *Arabidopsis* (*Wang et al., 2017*).

Subsequently, a number of DREB genes have been identified and characterized in a wide variety of plant species such as *Arabidopsis* (*Hwang et al., 2012*), rice (*Dubouzet et al., 2003*), bell pepper (*Hong & Kim, 2005*), soybean (*Chen et al., 2007*), pearl millet (*Agarwal et al., 2007*), wheat (*Egawa et al., 2006*; *Lucas et al., 2011* ), maize (*Feng et al., 2003*; *Liu et al., 2013*), chrysanthemum (*Yang et al., 2009*), tomato (*Guo & Wang, 2011*; *Hichri et al., 2016*), lettuce (*Park, Shi & Mou, 2020*). This depicts their involvement in different

abiotic stress responses. For instance, transgenic expression of DREB3a from a forage grass *Leymus chinensis* led to improved drought and salt tolerance in *Arabidopsis* (*Xianjun et al., 2011*). Another study reported enhanced drought and freezing stress tolerance by constitutive overexpression of *Arabidopsis* DREB1B gene in transgenic potato (*Movahedi et al., 2012*). Functional analysis of *Broussonetia papyrifera* DREB2 gene suggested its participation in drought and salinity stress responses. Furthermore, transgenic expression of BpDREB2 gene in *Arabidopsis* demonstrated enhanced salt and cold stress (*Sun et al., 2014*). Studies on *Medicago truncatula* revealed the role of DREB gene members both in freezing and cold stress (*Shu et al., 2016*). A novel DREB gene has been reported in maize namely ZmDBF3 which exhibits positive regulation relationship under salinity, drought, heat and cold stress (*Zhou et al., 2016*).

The distribution of DREB genes varies greatly between plants. Polyploidization is thought to have a major contribution in genome evolution and plant diversity (*Wendel et al., 2016*). Gene duplication is considered as a crucial mechanism in the evolutionary history of plants. Plant diversification is greatly aided by gene duplication events, which results in the generation of novel genes necessary for plant evolution (*McKain et al., 2016*). According to a recent study, polyploidization has played vital roles in the expansion of the DREB gene family specifically for plants that have undergone recent whole genome duplication (WGD) events such as t for Commelinid- specific and r for Poaceae (*Wang, Ma & Lin, 2019*). The DREB gene family has expanded significantly, allowing for more extensive and complex functional distinction. For instance, 30 DREB genes have been characterized in mung bean, five of which are highly expressed under drought stress (*Labbo et al., 2018*). In barley, 41 DREB genes have been discovered, many of which are expressed in drought and salt stress (*Guo et al., 2016*). Studies on *Musa acuminata* and *Musa balbisiana* revealed 81 and 99 DREB genes, respectively (*Lakhwani et al., 2016*). A large number of DREB genes (210) haven been identified in wheat genome. Over-expression of TaDREB3-AI displayed enhanced tolerance to heat, dehydration, and salinity stresses (*Niu et al., 2020*).

Potato, which originated from the Andean regions of Bolivia and Peru (*Davies et al., 2005*), is the third most significant agricultural crop worldwide after wheat and rice. Potato is highly adaptable to a wide range of ecosystems. According to FAO, over 388 million tons of potatoes were produced annually with consumption of over 239 million tons in 2017 (*Handayani, Gilani & Watanabe, 2019*). Being a wholesome food, potato is rich in vitamins, minerals and complex carbohydrates (*Hussain, 2016* ). However, potato is highly sensitive to various types of stress thus affecting its sustainable production. The key factors that severely affect potato's yield are abiotic stresses, including drought, low temperature, heat and salinity (*Dahal et al., 2019* ). In previous studies, characterization of two DREB genes (StDREB1 and StDREB2) showed a remarkable increase in their expression by salinity stress. The ectopic expression of these genes in potato conferred enhanced tolerance to salinity stress in transgenic potato lines (*Bouaziz et al., 2015b*; *Bouaziz et al., 2012*). Furthermore, overexpression of StDREB1 and StDREB2 led to enhanced drought tolerance in transgenic potato lines (*Bouaziz et al., 2015a*). In addition, overexpression of DREB transcription factors imparted increased cadmium (Cd) stress

tolerance in transgenic potato cultivars (*Charfeddine et al., 2017*). Despite overexpression, little is known about characterization of the DREB gene family in *S. tuberosum*. Given the vital role that DREB genes play in plant's abiotic stress tolerance, it is essential to identify and study the DREB gene family in the potato genome.

In the present study, we identified all potential DREB genes encoded in the *Solanum tuberosum* genome. Further, we performed bioinformatics analysis for classification of DREB genes into different subgroups, presence of characteristic motifs, exon/intron organization, chromosomal distribution, and gene duplication events. Finally, we analyzed homology of DREB genes with *A. thaliana* and *S. lycopersicum*, functional diversity, and subcellular localization of StDREB genes. The results gained herein will provide useful insights for future studies on functional characterization of DREBs in Potato.

# MATERIALS AND METHODS

## Retrieval and identification of DREB genes in *Solanum tuberosum* genome

To perform a comprehensive identification of the DREB gene family members in *S. tuberosum*, the amino acid sequences of *Arabidopsis* DREB proteins were retrieved from TAIR database (https://www.arabidopsis.org/) (*Huala et al., 2001*) and used as queries for BLASTp homology search against the Potato genome v4.03 with an e-value of $1 \times 10^{-5}$ in Sol Genomics Network (SGN) (https://solgenomics.net/) (*Mueller et al., 2005*) and Phytozome v12.1 database (https://phytozome.jgi.doe.gov/pz/portal.html) (*Goodstein et al., 2012*), respectively. Since DREB gene family consists of only one conserved AP2 domain, all retrieved amino acid sequences of potato were scanned for the presence of AP2 domain using PFAM (http://pfam.xfam.org/) (*Mistry et al., 2021*) and SMART (http://smart.embl-heidelberg.de/) (*Letunic & Bork, 2018*) programs. Furthermore, amino acid conservation analysis was performed by alignment of the AP2 domain for each filtered DREB protein using MEGA X. The candidate protein sequences were validated for the presence of valine (V) at position 14th and glutamic acid (E) at position 19th, especially valine, which has been a characteristic feature for DREB gene selection. Amino acid positions were also assessed by ScanProsite server (https://prosite.expasy.org/scanprosite/) (*Hulo et al., 2006*). Additionally, physiochemical properties of DREB proteins such as protein length, molecular weight (kDa) and isoelectric point (pI) were computed by ExPASy ProtParam tool (https://web.expasy.org/protparam/) (*Gasteiger et al., 2003*).

## Phylogenetic analysis StDREB proteins

Full length amino acid sequences of AtDREB and StDREB proteins were aligned in order to construct the phylogenetic trees. Multiple sequence alignment (MSA) was executed by MUSCLE algorithm with default parameters. Following alignment, phylogenetic trees were constructed by neighbor-joining (NJ) method based on pairwise deletion and Poisson substitution model with 1,000 bootstrap replicates using MEGA X software (http://www.megasoftware.net) (*Kumar et al., 2018*). Based on the classification

schemes for AtDREB proteins, recently identified StDREB proteins were characterized into distinct subgroups.

## Gene structure and protein motif features

Genomic and complete coding sequences (CDS) of each StDREB gene were downloaded from Phytozome v12.1 to analyze exon-intron structures. Gene Structure Display Server (GSDS 2.0) (http://gsds.gao-lab.org/) was utilized for a detailed graphical illustration of the exon-intron organization by comparing CDS sequences of the StDREB genes with their respective genomic sequences (*Guo et al., 2007*). The MEME version 5.3.2 (https://memesuite.org/meme/tools/meme) was employed to detect conserved motifs in *S. tuberosum* DREB proteins with the following parameters: (i) an optimal motif width of 6–50 amino acids, (ii) zero or one occurrence per sequence, and (iii) a maximum number of motifs set to 15 (*Bailey et al., 2015*).

## In silico chromosomal mapping, gene duplication events and synteny analysis

Chromosomal position information of each StDREB gene was determined from Phytozome v12.1 database. Physical locations and relative distances of the StDREB genes were mapped on to their respective potato chromosomes using MapChart 2.2 Software (https://www.wur.nl/en/show/Mapchart.htm) (*Voorrips, 2002*). To analyze gene duplication events two mechanisms of gene expansion were considered: tandem duplications and segmental duplications. Two gene pairs situated on same chromosomal fragment and segregated by five or fewer gene loci were regarded as tandem duplications. To assess the effect of selective pressure and divergence time of StDREB genes, the Ka (non-synonymous) and Ks (synonyms) values were computed using ToolKit Biologists Tools (TB tools) software (https://github.com/CJ-Chen/TBtools) (*Chen et al., 2020*). The approximate divergence time was estimated using the formula $T = Ks/2x^*MYA$, where $x = 6.56 \times 10^{-9}$ and $MYA = 10^{-6}$ (*He et al., 2016*). For synteny analysis, both the genomic and gff3 annotation files of *S. tuberosum*, *A. thaliana* and *S. lycopersicum* were extracted. Circos Plot and synteny images were constructed and visualized through advanced Circos and dual synteny plotter software (https://github.com/CJ-Chen/TBtools) to examine segmental duplication gene pairs and orthologous gene conservation of StDREBs with other plants species respectively.

## Gene Ontology (GO) annotation and subcellular localization prediction

Gene ontology annotation analysis of StDREB genes was conducted through Blast2Go software (https://www.blast2go.com/) (*Conesa & Götz, 2008*). Transcript sequence of each StDREB gene was uploaded to this software to determine biological processes (BP), cellular compartments (CC) and, molecular functions (MF). To execute Blast2Go functional annotation BLASTx search, InterPro Scan, mapping and annotation were carried out with default settings. The WoLF PSORT tool (https://wolfpsort.hgc.jp/) (*Horton et al., 2007*) and CELLO Online server v.2.5 (http://cello.life.nctu.edu.tw/) were used to predict the subcellular localization of DREB genes.

## RESULTS

### Identification and characterization of putative DREB genes in *S. tuberosum* genome

To identify all potential DREB gene family members in Potato, DREB protein sequences from the model plant *Arabidopsis thaliana* were used as queries in BLASTp homology search against *S. tuberosum* genome v4.03. A total of 66 DREB genes designated as StDREB1 to StDREB66 were identified in Potato by removing other redundant hits with insignificant e-value and identity percentage. Based on Phytozome v12.1 annotation, all of the StDREB genes belonged to dehydration responsive element binding protein and ethylene-response factor (Data S1). Detailed information of all the StDREB genes aligned with their respective *Arabidopsis* ortholog were predicted (Data S2). Later, StDREB genes were verified by using SMART and Pfam analysis in order to confirm the presence of one conserved AP2 domain as shown in Table 1 followed by amino acid conservation analysis. Valine has been indicated as the most significant amino acid for binding affinity whereas glutamic acid might have some flexibility among proteins. Among 66 StDREB proteins, 26 sequences presented only valine residue at the position 14th, while 40 sequences exhibited both V and E at position 14th and 19th within the conserved AP2 domain, respectively. StDREB1 to StDREB66 genes were subjected to further analysis in order to assess their physiochemical characteristics. Analysis depicted that the amino acid lengths of 66 StDREB proteins varied greatly from 72 a.a residues (StDREB51) to 457 a.a residues (StDREB31) with different range of molecular weights from 8.3 kDa to 52.29 kDa and isoelectric points (pI) ranged from 4.17 (StDREB59) to 9.97 (StDREB51), respectively as provided in Data S2.

### Phylogeny and group classification of StDREB genes

To investigate the evolutionary relationship and sequence homology between *Arabidopsis* and *S. tuberosum* DREB protein sequences, a Neighbor- Joining (NJ) phylogenetic method with 1,000 bootstrap replicates was used to generate a phylogenetic tree for DREB proteins among *S. tuberosum* (66 DREB proteins) and *A. thaliana* (56 DREB proteins) (Data S3). The phylogenetic tree was constructed followed by multiple sequence alignment (MSA) of AP2 domain coding protein sequences present in potato DREB sequences. All 66 genes of DREB family in potato were categorized into six subgroups referred to as A1, A2, A3, A4, A5 and A6 with reference to the classification of AtDREBs as shown in Fig. 1.

The largest subgroup A6 consisted of 19 members while the smallest subgroup A3 consisted of only two StDREB members. A total of 11, 8, 18 and 8 protein members were assigned to A1, A2, A4 and A5 subgroups, respectively.

The conserved amino acid sequence present in AP2 domain of StDREB members had valine at position 14th and glutamic acid at position 19th but some members only had valine which plays an important role in identification of the DREB gene family's various DNA binding sites. The phylogenetic tree revealed that StDREB members (StDREB50, StDREB52, StDREB39, StDREB40, StDREB56, StDREB65, StDREB61, StDREB66, and

**Table 1 Conserved AP2 domain sequences from six representative StDREB genes. AP2 domain sequences identified in StDREB genes by Pfam.** Different rows indicated alignment of the AP2 domain. First row depicted consensus region of the Hidden Markov Model (HMM); second row depicted the match between query protein sequences and HMM; third row demonstrated the degree of confidence in each individually aligned residue; whereas fourth row demonstrated the query sequence.

| Gene ID | PF00847 AP2 domain | | Sequence alignment | HMM length |
|---|---|---|---|---|
| StDREB3 | #HMM | pkikGVrydkkrgrWvAewsk.ngkrkkkrfsvgkyGf.eeAkraAiaarkkleg | 60–110 | 54 |
| | #MATCH | p+++GVr + +g+Wv+e ++ n+ k+r+++g++ + e+A+ra++ a+ +l+g | | |
| | #PP | 78********.5******999775...9**********************998 | | |
| | #SEQ | PVYRGVRMRN-SGKWVCEVREpNK—KTRIWLGTFPTaEMAARAHDVAAIALRG | | |
| StDREB12 | #HMM | ikGVrydkkrgrWvAewsk.ngkrkkkrfsvgkyGf.eeAkraAiaarkkleg | 18–66 | 54 |
| | #MATCH | ++G+r +k +g+WvAe ++ n+ + r+++g y + A+ra++ a ++l+g | | |
| | #PP | 9*******.9******999775...9*****************9998 | | |
| | #SEQ | YRGIRMRK-WGKWVAEIREpNK—RSRIWLGSYSSpVAAARAYDTAVFYLRG | | |
| StDREB13 | #HMM | pkikGVrydkkrgrWvAews | 69–87 | 54 |
| | #MATCH | +k++GVr++ +g+WvAe + | | |
| | #PP | 79********.9******65 | | |
| | #SEQ | CKYRGVRQRT-WGKWVAEIR | | |
| StDREB28 | #HMM | pkikGVrydkkrgrWvAewsk.ngkrkkkrfsvgkyGf.eeAkraAiaarkklege | 58–109 | 54 |
| | #MATCH | p + GVr++ +g+Wv e ++ + kk r+++g++ + e+A+ra++ a+ ++ g+ | | |
| | #PP | 679********.9******97744...69********************99986 | | |
| | #SEQ | PIYHGVRKRS-WGKWVSEIREpR—KKSRIWLGTFSTaEMAARAHDVAAIAIKGH | | |
| StDREB47 | #HMM | ikGVrydkkrgrWvAe...wskngkrkkkrfsvgkyGf.eeAkraAiaarkkleg | 184–232 | 54 |
| | #MATCH | ++GVr+++ +g+WvAe + ++ ++r ++g++++ e A++a+++ ++kl+g | | |
| | #PP | 9*******.9******5554..55...8******************9999987 | | |
| | #SEQ | YRGVRQRH-WGKWVAEirlP–RN—RTRLWLGTFDTaEDAAMAYDREAYKLRG | | |
| StDREB63 | #HMM | kikGVrydkkrgrWvAewsk.ngkrkkkrfsvgkyGf.eeAkraAiaark | 64–109 | 54 |
| | #MATCH | +++GVr++ +g+WvAe ++ + k++r ++g++ + e A+ra+++a+ | | |
| | #PP | 69********.9******87744...69*****************996 | | |
| | #SEQ | RYRGVRQRS-WGKWVAEIREpR—KRTRRWLGTFATaEDAARAYDRAAI | | |

StDREB62) belonging to group six exhibited greater homology with each other rather than with *Arabidopsis* DREB sequences due to high structure similarity among them which was further investigated by gene structural analysis and conserved motifs identification in potato.

## Gene structure and motif composition of StDREB members

The organization of introns and exons is pivotal in the evolution of gene families. Gene structure analysis was conducted through aligning the cDNA and gDNA sequences (Data S4) to gain further intuition into the structural similarity and divergence of DREB genes in Potato. We also constructed a phylogenetic tree as shown in Fig. 2 but using only full-length protein sequences of StDREB genes to relate it with exon/intron distribution and motif composition. Results showed that eight StDREB genes (21, 34, 60, 20, 13, 39, 40, and 52) had one intron with the exception of StDREB56 which contained three introns in its coding region while other 57 StDREB genes had only exons in their coding region as shown in Fig. 2B. MEME motif detection web software was utilized to recognize the conserved motifs to further understand the diversification in StDREB members of Potato Fig. 2C. We found fifteen conserved motifs with different amino acid

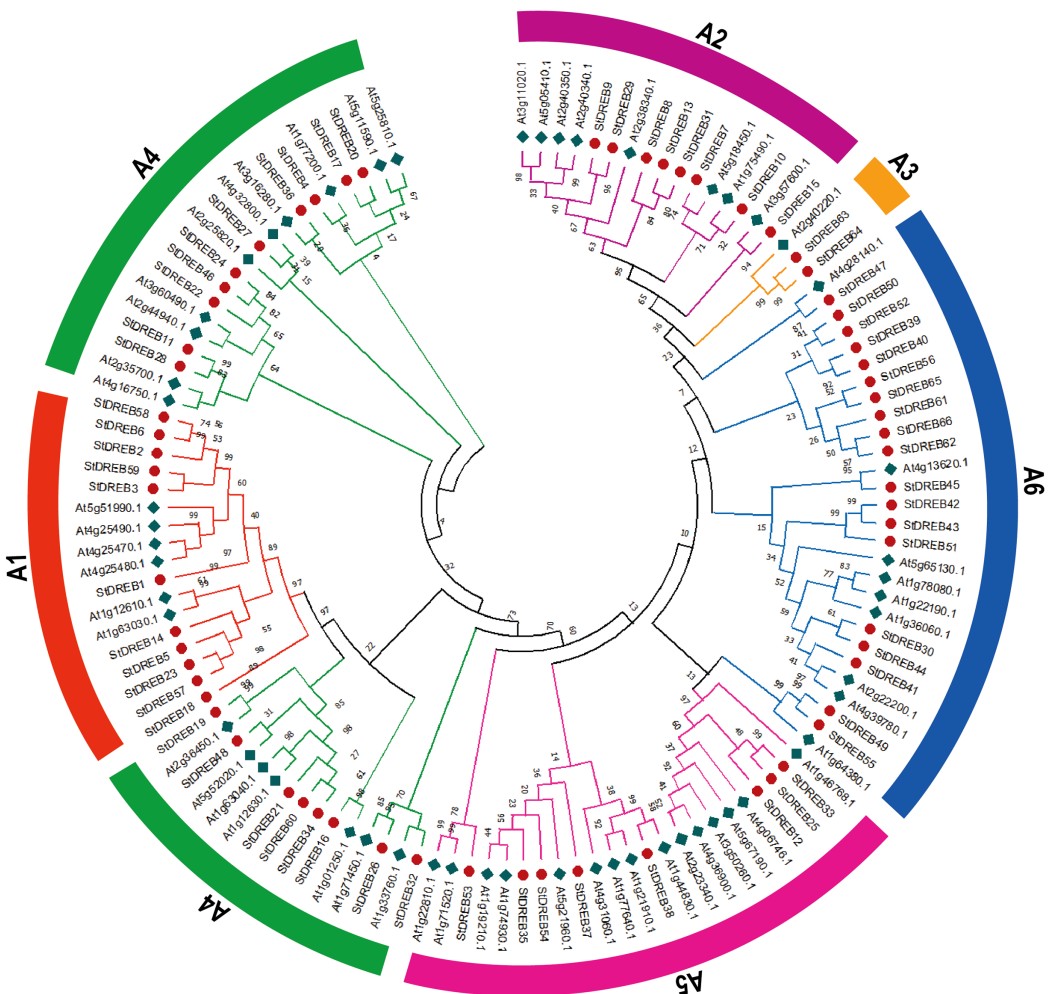

**Figure 1 Phylogenetic tree of DREB proteins among *Arabidopsis* and *S. tuberosum.*** StDREB proteins were assigned to six distinct subgroups (A1–A6) based on the classification of *Arabidopsis*. The six subgroups were highlighted with different colors.

range from 9–50. Motif 12 was the shortest with nine amino acid residues (DDDMSLWSY) while motif 15 was the longest with 50 amino acid residues (NNYJPYGFYPAVQYAEDISQNPQHSIQKQTFDDNYGFLDGETTKASGMIW). Motif composition in different subgroups was different (Figs. S1 and S2). Motifs 1 and 2 were found within the AP2 domain. Majority of the StDREB proteins observed motif 1 and motif 2 with very few exceptions. Motif 13 was unique to three StDREB (5, 23 and, 57) members belonging to A1 subgroup. Motif 15 was exclusively present in A3 subgroup (63, 64), whereas motif 8 was found both in A3 subgroup and seven out of eight (7, 9, 10, 13, 15, 29, and 31) members of A2 subgroup proposing that proteins in these subgroups may possibly share a specific function. Motif 9 was only encountered in five StDREB protein sequences (11, 22, 24, 28, and 46) of the A4 subgroup. Motif 10, motif 11, and motif 14 were detected only within some members of A6 subgroup. Majority of StDREB proteins within the same subgroup exhibited similar intron-exon structure and motif compositions which supported the phylogenetic analysis of DREB gene family, while the

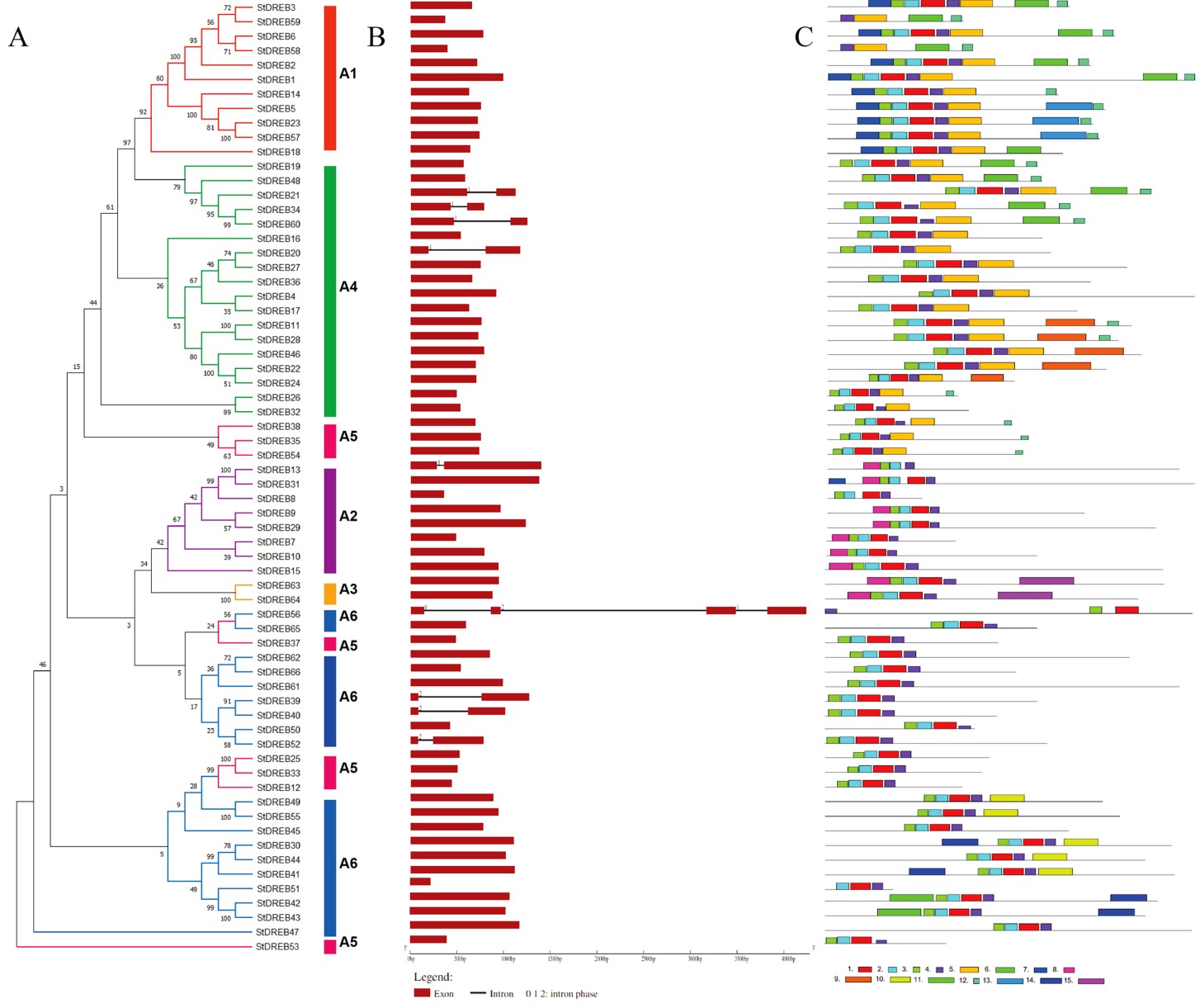

**Figure 2 Analysis of phylogenetic evolutionary relationship, intron-exon organization and protein motif patterns of StDREBs.** (A) A phylogenetic tree of 66 StDREB members was constructed with the neighbor-joining method using Mega X. The 66 StDREB proteins were then classified into six subgroups (A1–A6). (B) The gene structure was visualized using an online tool Gene Structure Display Server 2.0. The maroon boxes represented exons and the black line represented intron. The scale at the bottom showed the exon sizes. The numbers 0, 1, and 2 depicted the intron splicing phase. (C) Conserved motifs StDREB proteins were identified using MEME software. Fifteen predicted motifs were represented by distinct colored boxes and the grey lines indicated non-conserved regions.

difference among the distinct subgroups directed their diverse roles. The similarities and structural variations among these motifs can be studied further to provide new insights. Most StDREBs classified in the same subgroup generally had a similar motif composition and might have similar functions. Details of 15 conserved motifs are summarized in Data S5.

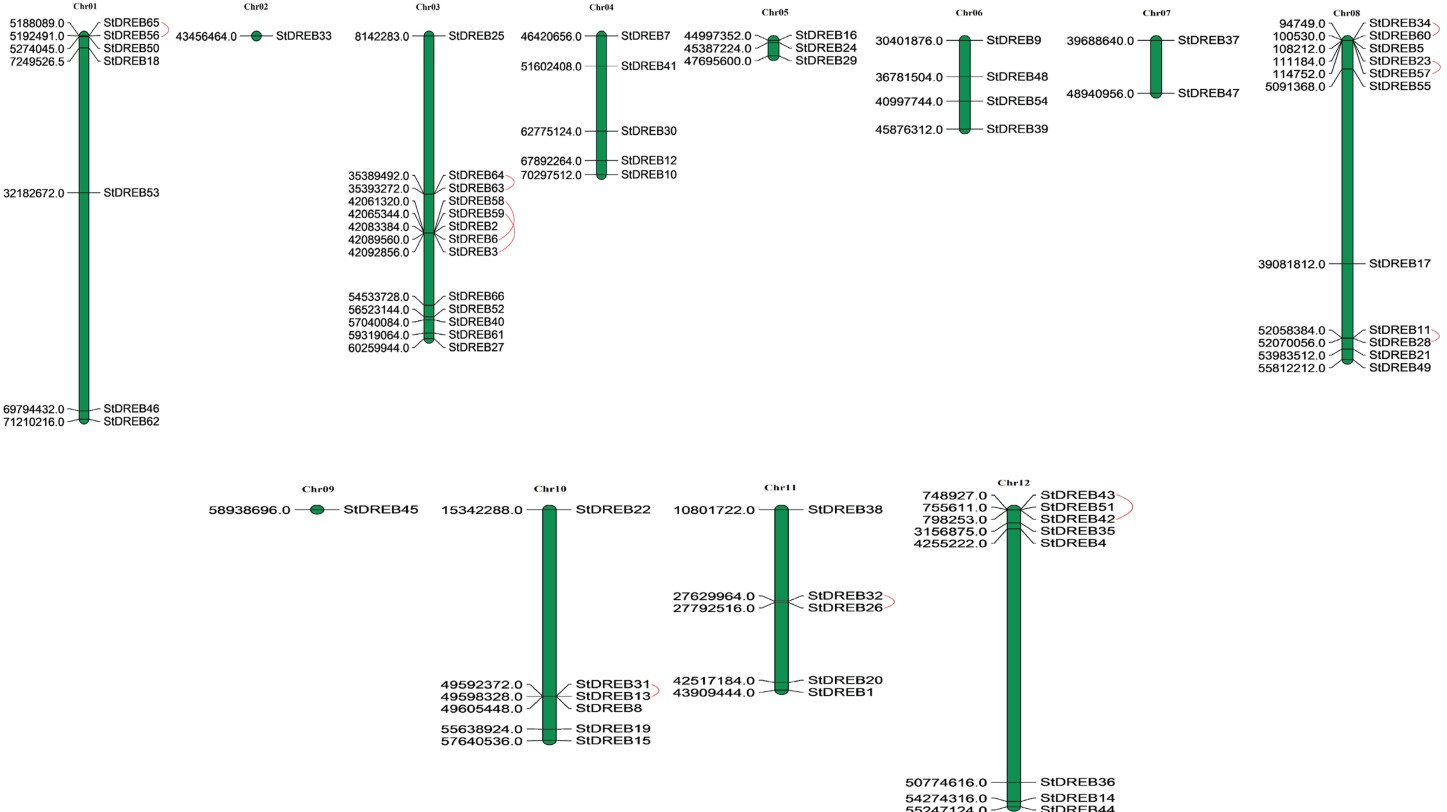

**Figure 3 Genomic distribution of 66 StDREB genes across 12 potato chromosomes.** Vertical bars indicated locus of StDREB genes on their respective chromosomes. The *x*-axis scale represented the chromosome length while tandem duplicated gene pairs harbored by various chromosomes were indicated in red.

## Chromosomal localization and gene duplication analysis of StDREBs

Chromosomal positions of all StDREB genes were obtained from Phytozome database v.12.1. Physical location of all the identified StDREB genes were mapped by using MapChart 2.2 software on to their corresponding chromosomes as depicted in Fig. 3. Chromosomal localization indicated that the 66 StDREB genes were heterogeneously distributed onto 12 chromosomes across the Potato genome which indicated that genetic variation occurred during the evolutionary process. Largest numbers of StDREB genes i.e., 13 were positioned on chromosome 3 (chr03). Conversely, chromosome 2 (chr02) and chromosome 9 (chr09) had the least number of StDREB genes i.e., 1. Two chromosomes (chr04 and chr11) hosted 5 StDREB genes. In addition, chromosome 8 (chr08) harbored 11 StDREB genes, chromosome 1 had 7 StDREB genes, chromosome 5 (chr05) and chromosome 7 (chr07) had 3 and 2 StDREB genes, respectively.

Tandem and segmental duplications contribute to the expansion of new gene family members and novel functions in the evolution of plant genome. To investigate gene duplication events within the potato genome, we investigated tandem and segmental duplications during evolution of StDREB gene family. A total of 10 StDREB gene pairs were confirmed to be tandem duplications. According to gene duplication analysis, it was

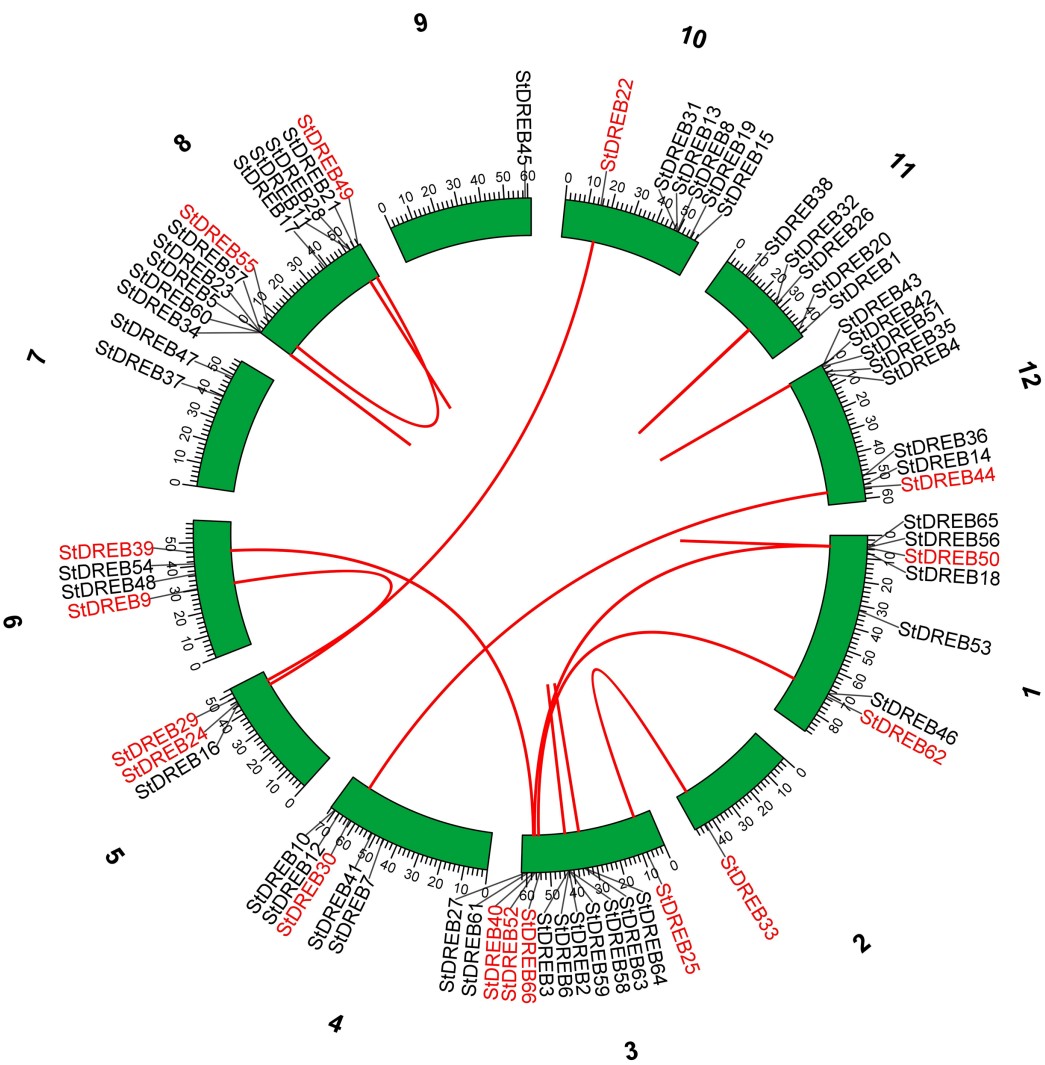

**Figure 4 Segmental gene duplication events exhibited by StDREB genes across 12 potato chromosomes.** Red lines represented segmental duplication of StDREB gene pairs. Chromosome number is located at top of each green block.               

observed that chr01, chr10, chr11 and chr12 each experienced one tandem duplicated gene pair while chr3 and chr8 harbored three tandem duplicated gene pairs illustrated as red lines in Fig. 3. Besides tandem duplication, 8 segmental duplication events were identified by constructing Circos Plot illustrated as red color in Fig. 4. To determine the selective evolutionary pressure on StDREB gene divergence after duplication, Ka and Ks values were computed for the duplicated StDREB gene pairs using KaKs calculator. Generally, KaKs = 1 implies neutral selection, Ka/Ks >1 implies positive selection and, Ka/Ks < 1 implies purification selection. Each duplicated StDREB gene pair had Ka/Ks < 1, which was indicative of purification selection during evolution. Furthermore, duplication events of 18 gene pairs were estimated to have occurred between 6.15 and 223.0 million years ago (Data S6).
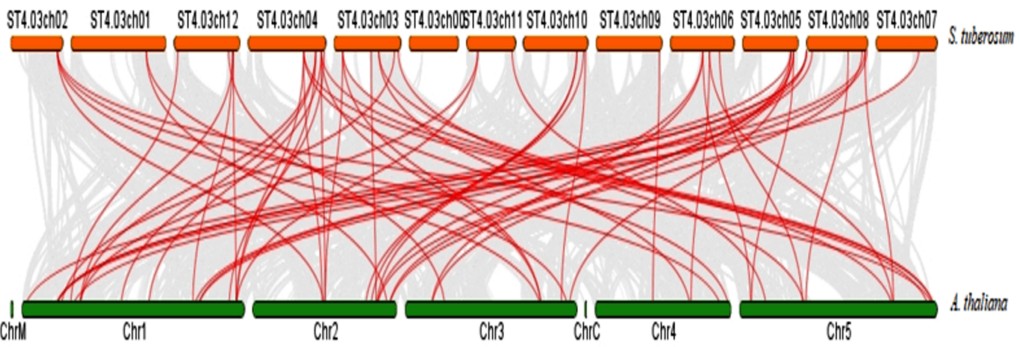

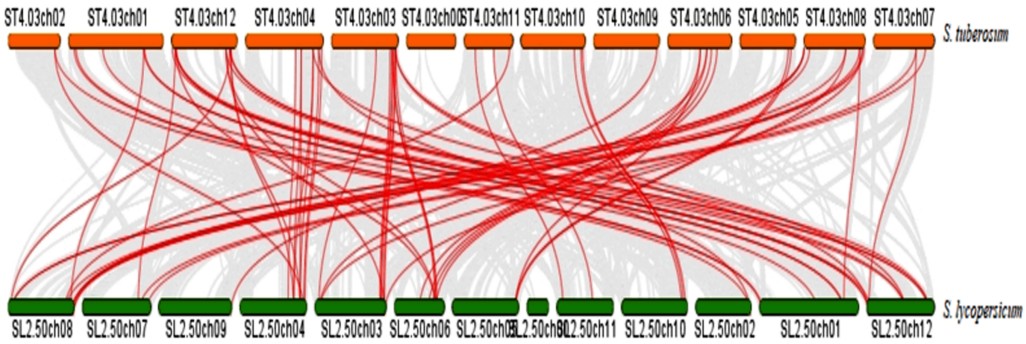

**Figure 5 Synteny relationship analysis of DREB genes between *S. tuberosum*, *A. thaliana* and *S. lycopersicum*.** Red and green bars represented the chromosomes while red lines indicated the homology and evolutionary link of StDREB genes with *A. thaliana* and *S. lycopersicum*, respectively. In addition, grey lines indicated all the collinear blocks present in their respective genomes.

## Synteny relationship of StDREB genes

To further infer the evolutionary relationship, DREB genes were compared to identify orthologous StDREB gene pairs between *S. tuberosum*, *A. thaliana* and *S. lycopersicum*. According to the synteny analysis, 36 of 66 StDREB genes had collinear gene pairs in *A. thaliana* and 45 of 66 StDREB genes had corresponding orthologs in *S. lycopersicum*, respectively as provided in Data S7. The syntenic maps showed high evolutionary homology relationship of StDREB genes with *A. thaliana* and *S. lycopersicum* DREB genes implying that they might have related functions as depicted in Fig. 5.

## Gene Ontology (GO) annotation and subcellular localization prediction of StDREB genes

To explore the functions of StDREB genes in different biological processes, molecular functions and cellular compartment building, GO functional annotation was performed as depicted in Fig. 6 and gene ontology number was also identified during the analysis as provided in Data S8. Biological processes showed that the StDREB genes were involved in cell metabolism, defense response and ethylene activated signaling. Majority of the StDREB genes were involved in sequence specific DNA-binding functions and
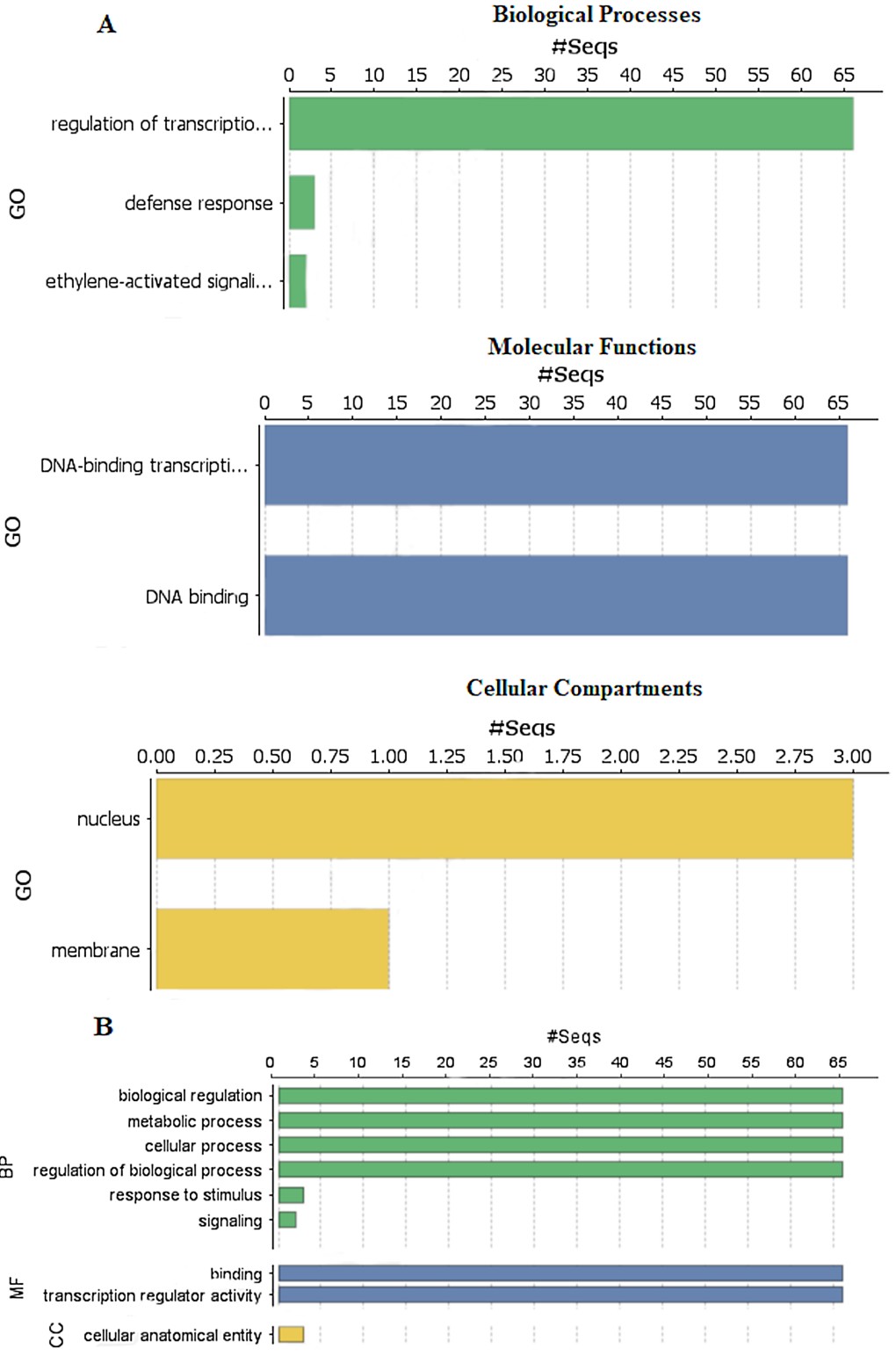

**Figure 6 Gene Ontology (GO) and functional enrichment analysis of StDREB genes by using Blast2GO software.** (A) GO distribution by level processes. (B) Direct count GO. Green bars represented biological processes (BP), blue bars represented molecular functions (MF), and yellow bars represented cellular compartments (CC).

transcriptional regulation, indicating their significance as a transcription factor for potato growth and development, as well as regulating the expression of abiotic stress responsive genes. The cellular compartment study revealed that most of the StDREB genes were concentrated in cell nucleus with the exception of StDREB47 and StDREB53 which were localized in cytoplasm. In addition, some StDREB genes were found in cell membrane to control the signal transduction inside the cell.

To predict the subcellular localization of StDREB proteins, two different tools were employed. Results confirmed that StDREB proteins were predominantly localized in the nucleus followed by mitochondria and chloroplast, respectively as indicated in the heat map (Fig. 7; Data S9).

## DISCUSSION

Potato is one of the world's most important crops, providing food to over 100 countries. Potato is vulnerable to several abiotic stresses, and its steady development is jeopardized by recurrent stress outbreaks (*Mirzaei, Bahramnejad & Fatemi, 2020*). The DREB gene family plays an important role in plant's abiotic stress signaling. DREB gene family is well known for its highly conserved AP2 domain that can specifically bind to the DRE/CRT cis-acting elements to activate expression of several stress tolerance genes, thereby enhancing plant tolerance (*Chen et al., 2016b*). With the availability of whole-genome sequence, members of the DREB gene family have been identified in many plant species. Until now, bioinformatic analysis of DREB gene family has not been reported in potato. Therefore, our current study furnished a detailed data on genome-wide characterization of DREB genes in *Solanum tuberosum*. By using *Arabidopsis thaliana* DREB genes as query sequences, we identified 66 putative DREB genes in *S. tuberosum* (Data S2) and categorized them into six subgroups corresponding to *A. thaliana* DREB orthologs. The amino acid conservation (valine at 14th and glutamic acid at 19th) confirmed the presence of DREB genes in *S. tuberosum* genome.

The genome-wide analysis revealed the difference in number of genes and genome size of *S. tuberosum* and *A. thaliana*. The identified number of DREB genes in *S. tuberosum* (66) is greater than that in *Arabidopsis* (56) which may be due to the difference in their genome sizes. When compared to previous studies, greater numbers of DREB genes were reported in *S. spontaneum* (110) (*Huang et al., 2020*), *P. trichocarpa* (75) (*Chen et al., 2013*), soybean (73) (*Zhou et al., 2020*), and malus (68) (*Zhao et al., 2012*), but less DREB genes were found in sesame (41) (*Dossa et al., 2016*), maize (51) (*Du et al., 2014*), mulberry (30) (*Liu et al., 2015*), grape (38) (*Zhao et al., 2014*), common bean (54) (*Konzen et al., 2019*), *Phyllostachys edulis* (27), and pineapple (20) (*Chai et al., 2020*). In order to comprehend the potential functions of proteins, it is inevitable to compute the physiochemical characteristics of plant protein families (*Salih et al., 2019*). The DREB proteins in potato demonstrated diversification in terms of physiochemical properties, which indicated that these genes might play various roles in plant development and defense response. The lengths of all DREB proteins had a range of 72–457 amino acid residues. The molecular mass largely varied between 8.3 and 52.29 kDa, and the isoelectric point ranged from 4.17 to 9.97 (Data S2). Moreover, our phylogenetic analysis of

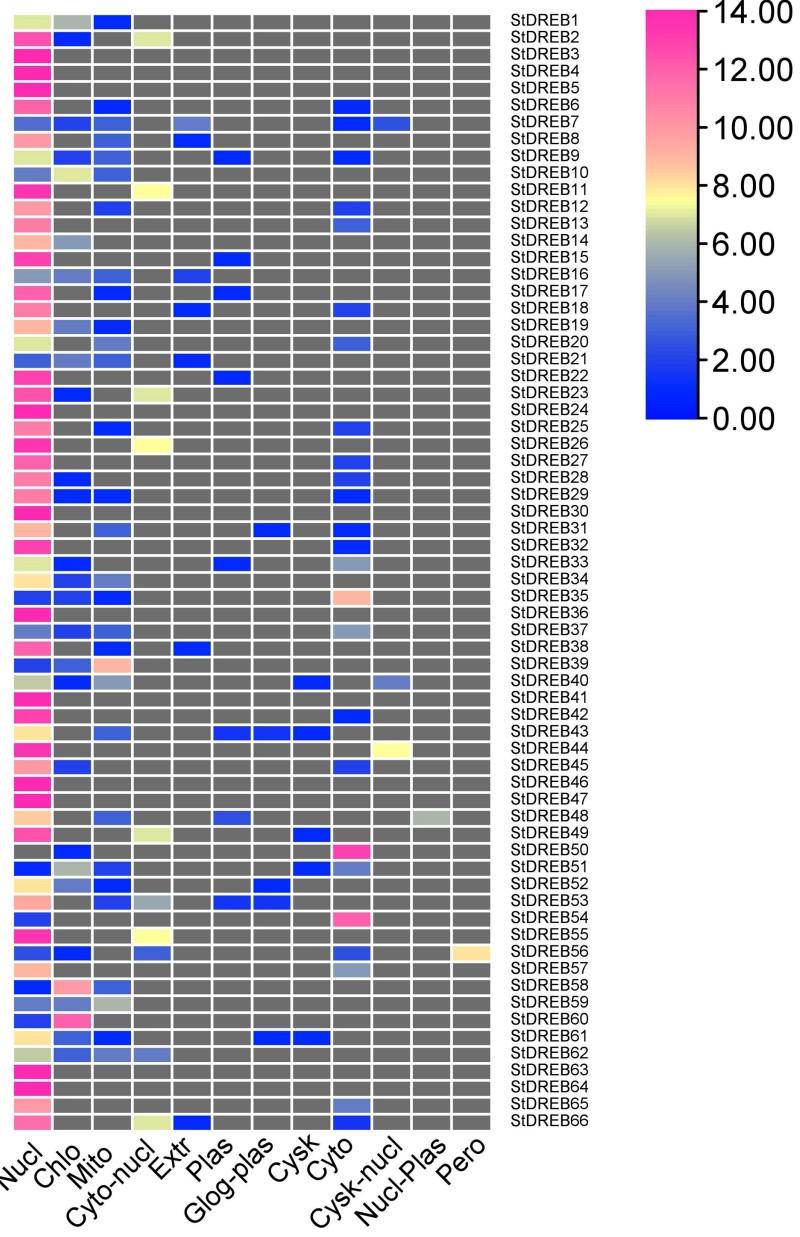

**Figure 7 Heat map of subcellular localization predictions for 66 StDREB proteins.** The color scale represents abundance of StDREBs in various cellular compartments.

*S. tuberosum* and *A. thaliana* DREB proteins demonstrated that the six subgroups (A1–A6) identified previously in *Arabidopsis* (*Akhtar et al., 2012*) were also present in *S. tuberosum*. The total number of genes for each group was 11, 8, 2, 18, 8, and 19, corresponding to the A1–A6 subgroups. This distribution was quite different from *A. thaliana* due to different genetic patterns in both plants (Fig. 1). Gene number similar to StDREB A3 subgroup had been reported in *Salix arbutifolia* (2) (*Rao et al., 2015*) and *P. trichocarpa* (2). However, 18 DREB genes were categorized in A3 subgroup of foxtail

millet (*Shi et al., 2018*), five in *S. spontaneum*, one in mulberry, one in *Phaseolus vulgaris*, and zero in pineapple.

To gain insights into the structural diversity of the StDREB genes, their gene structures were analyzed. Formerly, it was thought that DREB genes only contain coding region exons, without any intron as in *Arabidopsis*. But, later studies revealed that DREB genes contain both the exon and intron in their gene structure. This finding was further confirmed by gene structure analysis in wheat (*Sazegari & Niazi, 2012*), rice (*Matsukura et al., 2010*) and maize. Our study demonstrated nine StDREB genes consist of intronic regions along with exons. Out of nine StDREB genes, StDREB56 contain three introns whereas remaining 8 genes contain one intron only. While 76% of the StDREB genes showed a single exon (Fig. 2B). A previous study conducted on pineapple had also demonstrated highest number of introns as three in one of its 20 DREB genes. There was a strong connection of exon/intron structure between *S. tuberosum* and other species (*S. spontaneum*, soybean, pineapple, and populas) due to the presence of both exon and intron. Also, some studies showed that a compact gene structure with few or no introns enhanced timely response to various abiotic stresses in plants (*Jeffares, Penkett & Bähler, 2008*). The variation found in gene structure of StDREB genes elucidates the functional diversification which might be due to climatic or evolutionary changes in the plant genome profile (*Li et al., 2020*). Recent studies revealed the presence of some conserved motif sequences in transcription factors of *F. tataricum* located on the same chromosome (*Liu et al., 2019*), similar profile was observed in *A. thaliana* (*Sakuma et al., 2002*) which may be due to polyploid changes that occurred in the genome. Same results were obtained in *S. tuberosum* DREB genes through identification of conserved motif sequences. We identified 15 conserved motifs in StDREB genes along with the conserved AP2 domain (Fig. 2C). The AP2 domain has one alpha helix and three beta sheets at the N-terminus (*Wang et al., 2011*). The identified motifs in *S. tuberosum* DREB genes indicated high similarity with AP2 structure pattern. All conserved motifs had varied composition (Figs. S1 and S2). Transcription factor domains and motifs are frequently linked to DNA binding, transcriptional activity, and protein interaction (*Liu, White & MacRae, 1999*). The gene structure and motif analysis of the same subgroup were alike, thus validating the reliability of the phylogenetic tree classification. This finding was in line with the previous studies on DREB, which found that broad similarities existed in motifs and exon/intron structure between members of the same subgroup.

Chromosomal positions demonstrated uneven scattering of 66 DREB genes over 12 potato chromosomes. The asymmetrical arrangement of genes has been suggested to reveal information about their evolution (*Chen et al., 2016a*). Whole genome duplications (WGD) or polyploidization have been considered as the major causes of evolution that gives rise to novel traits and new transcriptional regulatory sites that can alter expression patterns (*Panchy, Lehti-Shiu & Shiu, 2016*). In total, 10 homologous pairs were confirmed to be produced by tandem duplication events (Fig. 3). In contrast, eight paralogous gene pairs were produced through segmental duplication events (Fig. 4). Moreover, gene pairs with tandem duplications were found on chromosomes of the same origin. The most recent tandem duplication events were estimated to be six million years ago for

two pairs of genes belonging to A3 and A6 subgroup, respectively (Data S6). *S. spontaneum*, *P. trichocarpa*, and soybean genome had undergone both tandem and segmental duplications whereas only tandem duplications were found in *P. vulgaris*. Tandem duplications are also known to be adaptively important in the development and function of abiotic stress responsive genes. Previous studies reported that tandem repeats often share common cis-acting elements, and may perform similar functions (*Flagel & Wendel, 2009*). Hence, our study also emphasizes that StDREB tandem gene duplication pairs may share similar functions and regulatory elements in their promoter region. To further investigate the potential evolutionary mechanisms of the *S. tuberosum* DREB gene family, interspecies synteny was inferred. The number of orthologous events of StDREB genes was greater with SlDREB genes as compared to AtDREB genes, which may be due to close evolutionary link between *S. tuberosum* and *S. lycopersicum* (Fig. 5). GO enrichment has been considered as a powerful tool for enhancing the understanding of functional genomics and underlying molecular mechanism. GO annotation verified the DNA-binding ability of StDREB genes which was consistent with the findings in *P. vulgaris* (Fig. 6). All the StDREB genes were concentrated inside the nucleus. Furthermore, our results from the subcellular localization also indicated that StDREB proteins were primarily present inside the nucleus (Fig. 7).

## CONCLUSIONS

In conclusion, we performed genome-wide identification and characterization of the DREB gene family in *S. tuberosum* and conducted a detailed investigation of their evolutionary relationship, genome organization, duplication events, and functional annotation using bioinformatics tools. In total, 66 DREB genes having an AP2 domain and conserved amino acid residues (Valine at position 14th) were identified in the Potato genome and unevenly mapped across 12 chromosomes. Based on the sequence alignment and phylogenetic analysis, StDREB genes were classified into six subgroups (A1–A6) corresponding to previous report of *Arabidopsis*. The results of gene structure and conserved motif analysis were found consistent with the phylogenetic classification. Gene structures and motif patterns showed that the StDREB members in the same subgroup displayed broad similarities. The expansion of the DREB gene family in Potato was aided greatly by tandem and segmental duplications. Evolutionary divergence analysis (Ka/Ks) suggested that the StDREBs were under strong purification selection during plant evolution. Synteny relationship analysis indicated that 35 and 46 StDREB genes were orthologous to *Arabidopsis* and *S. lycopersicum*, respectively. Furthermore, subcellular localization revealed that the StDREB genes were primarily located inside the nucleus. Through gene ontology (GO) annotation, we found most of the StDREB genes had DNA binding function, which suggested their role as important transcriptional activators. Functional enrichment indicated pivotal roles of StDREB genes in cell development, defense responses, and hormone signaling. Overall, our data delineated the evolutionary characteristics and genome duplication events along with biological and molecular functions of DREB genes in *S. tuberosum*. Taken together, our results will

provide a foundation for unraveling the molecular mechanisms and further functional characterization of the StDREB gene family, thus providing sources for plant breeding and genetic engineering.

## ABBREVIATIONS

| | |
|---|---|
| **DRE** | dehydration responsive element |
| **CRT** | C-repeat |
| **AP2** | APETALA2 |

### Funding
The authors received no funding for this work.

### Competing Interests
The authors declare that they have no competing interests.

### Author Contributions
- Nida Mushtaq performed the experiments, analyzed the data, prepared figures and/or tables, and approved the final draft.
- Faiza Munir conceived and designed the experiments, analyzed the data, prepared figures and/or tables, authored or reviewed drafts of the paper, and approved the final draft.
- Alvina Gul analyzed the data, authored or reviewed drafts of the paper, and approved the final draft.
- Rabia Amir analyzed the data, authored or reviewed drafts of the paper, and approved the final draft.
- Rehan Zafar Paracha analyzed the data, authored or reviewed drafts of the paper, and approved the final draft.

### Data Availability
   The raw data is available in the Supplemental Files.

### Supplemental Information
Supplemental information for this article can be found online at http://dx.doi.org/10.7717/peerj.11647#supplemental-information.

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
