# Peer review of "Genome-wide analysis, identification, evolution and genomic organization of dehydration responsive element-binding (DREB) gene family in Solanum tuberosum"

_PeerJ, doi:10.7717/peerj.11647_

## Round 0.1 · original submission · Minor Revisions

We have received 3 detailed reviews on your manuscript. All the reviewers suggested minor revision. Please consider the remarks.

·

Basic reporting

1. English language and manuscript write-up is of sufficient quality.
2. The authors provided up to date literature according to the background of the study
3. The MS is written and organized well. All the figures and tables are of good quality and understandable. The raw is shared and provided in supplementary files
4. No comment

Experimental design

1. The paper fall in aim and scope of journal
2. The research question is clear and authors identified the knowledge gaps.
4. No comment
5. Methods are described very well.

Validity of the findings

1. The research have novelty as very few findings published in this area.
2. All the data is analyzed statistically very well
3. The conclusion can be improved more
4. no comment

Additional comments

The work of Mushtaq and colleagues presents interesting results on Genome-wide analysis, identification, evolution and genomic organization of Dehydration Responsive Element Binding (DREB) gene family in Solanum tuberosum, and something important to highlight, is the importance of DREB gene family in abiotic stresses.
I couldn´t find any limitations. Perhaps the authors limit the role of DREB in abiotic stresses. The strengths of the work are the novelty of results, bioinformatics, good introduction, and in general, the ms. is well written

Comments
1. In abstract line 35 ‘S. tuberosum’ should be Solanum tuberosum’ and line 46’ S. lycopersicum’ should be ‘Solanum lycopersicum’
2. In abstract section line 40 word ‘strengthened’ to be replaced by ‘Supported’
3. The concluding sentence, line 50, can be written like this, this study provides a comprehensive and systematic understanding of precise molecular mechanism and functional characterization of StDREB genes in abiotic stress responses and will lead to improvement in S. tuberosum
4. The introduction can be written more better like include some sentences on gene duplication and evolution.
5. In result section in sentences structure ‘the’ is use too many times. Remove unnecessary ‘the’ from sentences.
6. In discussion section few more (2-3) relevant references aligned with the subject study can be added.
7. In Table 1, the different colors represent what ? Indicate it to make it more meaningful
8. English language and manuscript write-up is of sufficient quality. The quality of figures is appropriate.
9. The data underlying the study is significantly provided in article and supplementary files.
10. The manuscript has a good correlation between the objectives, methodology and results.

Reviewer 2 ·

Basic reporting

Clear, unambiguous, professional English
language used throughout

Intro need to be improved&.
Literature well referenced & relevant.

Structure conforms to PeerJ standards,
discipline norm, or improved for clarity.

Figures are relevant, high quality, well
labelled & described.

Raw data supplied

Experimental design

Original primary research within Scope of
the journal.

Research question well defined, relevant
& meaningful. It is stated how the
research fills an identified knowledge gap.

Rigorous investigation performed to a
high technical & ethical standard.

Methods described with sufficient detail &
information to replicate.

Validity of the findings

There are literature present on DREB but rare in S. tuberosume

All underlying data have been provided;
they are robust, statistically sound, &
controlled.

Conclusions are well stated and can be improved more, linked to
original research question & limited to
supporting results

Additional comments

The study has worked out the Genome-wide analysis, identification, evolution and genomic organization of Dehydration Responsive Element Binding (DREB) gene family in Solanum tuberosum. The work was taken up in StDREB gene family was comprehensively analyzed using bioinformatics approaches. The study is of general interest and not providing any novel information which can be exploited in crop improvement.
Comment 1: I have concern that did the studied was taken under any abiotic stress ?
Comment 2: A lot of literature is present on DREB family so please update literature throughout.
Comment 3: Do you perform identification of cis-acting regulatory elements in StDREB promoters?
Comment 4: Did you include the transcriptional level of the StDREB Genes? If yes then provide if not then why?
Comment 5: Did you compare the DREB family composition between S. tuberosum and other species? If not then why?
Comment 6: The discussion can be improved more.
Comment 7: Check English spellings and sentences throughout

Overall conclusion: The paper is well drafted and I will accept it after incorporation of these modification

Reviewer 3 ·

Basic reporting

The article is well drafted, clear and professional English used throughout

The references need to be updated

The article is organized well and raw data is also provided

No comment

Experimental design

The article falls in aim and scope of the journal
The research question is well defin3ed, relevant and meaningful. The knowledge gaps are already identified
The research is performed under high technical and ethical standard

The methods are described well self explanatory

Validity of the findings

The literature is already present on DREB family in different crops but scarce in potato
All data has been provided and is robust, statistically sound and controlled
Conclusion need improvement
No comment

Additional comments

In this article the authors comprehensively analyzed StDREB gene family using bioinformatics approaches which is quite scare in potato crop. The article is well drafted and organized, however I have few comments to improve the article more understandable for readers. Abstract is well written. In introduction the last paragraph need drastic improvement (Line 123-130). The same sentences were repeated many times. A lot of information is present on DREB family, so update your references accordingly. Materials and methods are well drafted. Results are also written well. However discussion need improvement. Do not repeated the sentences again in the discussion. Conclusion also need improvement and only report your main findings.

---

## Round 0.2 · accepted · Accept

Thanks for the update and detailed answer to the reviewers. All 3 of the reviewers have no more remarks now.

·

Basic reporting

Article is okay now

Experimental design

Article is okay now

Validity of the findings

Article is okay now

Additional comments

All the comments were incorporated by the authors and I have no other comments

Reviewer 2 ·

Basic reporting

Improved by the authors

Experimental design

No comment

Validity of the findings

Improved by the authors

Additional comments

I read the manuscript after the addition of comments and I have no comments. The article is now good

Reviewer 3 ·

Basic reporting

No more comments. The article is substantially improved

Experimental design

No more comments. The article is substantially improved

Validity of the findings

No more comments. The article is substantially improved

Additional comments

The article is substantially improved and I have no more comments. Now it is acceptable for publication